# Assessment of Clicker Training for Shelter Cats

**DOI:** 10.3390/ani7100073

**Published:** 2017-09-22

**Authors:** Lori Kogan, Cheryl Kolus, Regina Schoenfeld-Tacher

**Affiliations:** 1Department of Clinical Sciences, College of Veterinary Medicine and Biomedical Sciences, Colorado State University, Fort Collins, CO 80523-1601, USA; 2Clicker Learning Institute for Cats and Kittens, 2321 E Mulberry St, # 7 Fort Collins, CO 80524, USA; ckolus@gmail.com; 3Department of Molecular Biomedical Sciences, College of Veterinary Medicine, North Carolina State University, Raleigh, NC 27607, USA; regina_schoenfeld@ncsu.edu

**Keywords:** cats, animal shelter, behavior, environmental enrichment, clicker training, animal welfare

## Abstract

**Simple Summary:**

Living conditions in animal shelters can be stressful for cats. Clicker training might be able to alleviate this stress, by giving cats an opportunity to learn new behaviors and interact with humans. In this study, we assessed the initial ability of 100 shelter cats to perform four cued behaviors: touching a target, sitting, spinning, and giving a high-five. Each cat completed 15, five-min training sessions over a two-week span. At the end of the program, we assessed the cats’ ability to perform the same behaviors. On average, the cats performed better on all four behaviors after clicker training, suggesting that the cats could learn to perform specific behaviors on cue. Individual cats with a higher level of interest in food showed greater gains in learning for two of the behaviors (high-five and touching a target). Cats with a bolder temperament at post-assessment demonstrated greater gains in learning than those classified as shy. We suggest that clicker training can be used to enhance cats’ well-being while they are housed in shelters, and that the learned behaviors might make them more desirable to adopters.

**Abstract:**

Clicker training has the potential to mitigate stress among shelter cats by providing environmental enrichment and human interaction. This study assessed the ability of cats housed in a shelter-like setting to learn new behaviors via clicker training in a limited amount of time. One hundred shelter cats were enrolled in the study. Their baseline ability to perform four specific behaviors touching a target, sitting, spinning, and giving a high-five was assessed, before exposing them to 15, five-min clicker training sessions, followed by a post-training assessment. Significant gains in performance scores were found for all four cued behaviors after training (*p* = 0.001). A cat’s age and sex did not have any effect on successful learning, but increased food motivation was correlated with greater gains in learning for two of the cued behaviors: high-five and targeting. Temperament also correlated with learning, as bolder cats at post assessment demonstrated greater gains in performance scores than shyer ones. Over the course of this study, 79% of cats mastered the ability to touch a target, 27% mastered sitting, 60% mastered spinning, and 31% mastered high-fiving. Aside from the ability to influence the cats’ well-being, clicker training also has the potential to make cats more desirable to adopters.

## 1. Introduction

Cats and humans have a long history together, with the first relationships occurring approximately 10,000 years ago [1]. According to the 2015–2016 American Pet Products Association (APPA) survey, there are now approximately 85.8 million owned cats in the United States (compared with 77.8 million dogs), resulting in about 35% of all U.S. households having at least one cat [2]. Fifty-six percent of cat owners consider their cats to be family members and 41.5% consider them pets or companions; only 2.4% think of their cats as property. The most common place to acquire a cat is from a shelter or rescue, with 46% of cat owners reporting having obtained their cat from one of these organizations. This figure has increased from 43% in 2012–2013 [2]. Despite the growing number of cat owners obtaining their cats from a shelter, there are many more cats in shelters than homes available. As of March 2017, it was estimated that 3.2 million cats enter U.S. animal shelters each year, and approximately 70% of these cats are euthanized [2]. Furthermore, even in the best of shelters, the conditions are far from ideal for most cats. For example, decreased physical activity and a lack of environmental control can be challenging for shelter cats [3].

Regardless of whether a cat was found roaming outdoors or was an indoor owned cat, the shelter environment is novel and confining, leading to stress for many cats [4]. Although many shelters meet or exceed standards of care for physical health, most shelters offer minimal environmental enrichment [4]. Welfare and environmental enrichment for animals in shelters have recently received increased attention, with many people now recognizing that addressing an animal’s physical needs without considering its behavioral, social, or emotional needs is no longer adequate. Assessment of welfare in animal shelters, especially in those that house animals over longer periods of time, is of growing interest [5,6]. It is vital to ensure mental health is prioritized in shelter cats for several reasons, not least of which is to improve their chances of adoption. As a result, providing enrichment to improve animal welfare has received increasing focus, with research aimed at helping domestic cats cope with confinement [3].

Animal welfare has been defined as how an animal is coping within its living conditions. An animal is in a good state of welfare if it is healthy, comfortable, well-nourished, safe, able to express innate behaviors, and not suffering from pain, fear, or distress [7]. Psychological stress has been associated with a host of negative effects. In cats, physiological stress responses may include tachycardia, increased blood pressure, and elevated cortisol [8]. Stress can also cause behavioral changes such as increased hiding or decreased food intake and social interactions [9,10].

It has been suggested that environmental enrichment may have some mitigating effects on stress-related behaviors [11]. Common definitions of environmental enrichment refer to any physical, social, design, or management features that improve the environment of captive animals [12]. Using cats’ natural instinct to work for their food by teaching them new behaviors is one example of environmental enrichment [13]. Teaching a behavior with the use of a clicker device is one way to implement this type of enrichment. Despite the lack of prior research on domestic cat cognition, the use of clicker training as a form of enrichment can offer new opportunities and potentially positive welfare implications [14].

Clicker training, endorsed by the Humane Society of the United States [15], is based on Burrhus Frederic Skinner’s theory of operant conditioning that proposes that animals learn based on the consequences of their behaviors. Behaviors that are followed immediately by a desirable consequence (positive reinforcement) are more likely to occur again. Yet, even very brief delays in the delivery of consequences have been found to impair the rate at which animals learn to perform novel behaviors [16].

Clicker training uses an immediate signal while a desired behavior is being performed, which acts as a bridge to the forthcoming reward. This immediate, reward-predicting signal appears beneficial in supporting learning in situations where timely primary reinforcement is not feasible. As a result, many animal trainers have adopted such a reward-predicting signal. This technique was popularized as “clicker training” by Karen Pryor [17]. Clicker training employs a hand-held device that makes a clicking sound when pressed. The clicker is pressed when a desired behavior occurs and is typically followed by presentation of a food reward within a second or two [17]. Animal trainers report that the use of a clicker helps animals learn new tasks more quickly, as evidenced in several studies with different species [18]. Additionally, the use of positive reinforcement for training cats has been found to be more effective than forceful training techniques involving coercion [19]. In addition to enhancing learning, this type of training also allows for predictable interactions, thereby increasing an animal’s sense of control and the predictability of its environment, and as a result, its well-being and welfare [20,21].

Because of the potential benefits of clicker training, this study was designed to assess the ability of cats housed in a shelter-like setting to learn specific new behaviors in a limited amount of time using this modality. The study was approved by the regulatory compliance committee at Colorado State University.

Our primary hypothesis was that it is possible to train cats to perform particular behaviors in a relatively short time in a shelter environment through the use of clicker training. We did not expect that sex would have a determining effect on successful training. We did hypothesize that a cat’s level of interest in food and its temperament (shy vs. bold) would affect trainability, with cats that were less interested in food or that exhibited timid behavior (defined as unwilling to leave its cage) being less likely to be successfully trained.

## 2. Materials and Methods

All cats in this study were randomly chosen from a population of healthy, adoptable cats at least six months old that were residing in a limited admission, adoption-guarantee cat shelter. All cats were neutered and had variable lengths of stay (typically a few days to a few months) at the shelter prior to training. Cats were obtained from a variety of circumstances: strays, friendly community cats, owner surrenders, or transferred in from other shelters. During the study period, the selected cats were temporarily housed at a shelter-like facility (Clicker Learning Institute for Cats and Kittens (CLICK), a separate non-profit organization in Fort Collins, Colorado, USA) located in the same building as the cat shelter (but within its own unit). One hundred cats (57 (57%) females; 43 (43%) males) that completed a two-week clicker training program between 1 August 2016 and 28 April 2017 were assessed. Nineteen cats were removed from the study because they were adopted, became ill, or did not complete the requisite number of training sessions for other reasons.

While in the training program, cats were individually housed in cages at CLICK for two weeks. Inside dimensions of the cages were 88.9 × 68.6 × 73.7 cm (35 × 27 × 29 in) with built-in L-shaped shelves, three solid walls and vertically barred doors facing the interior room. All cages were along one wall so cats were not facing other caged cats. Cages were enriched with soft bedding, multiple toys, and cardboard scratchers in addition to litter boxes and food and water dishes. If cats appeared fearful, they were provided a cardboard box to hide in. Cats 6.8 kg (15 lbs) or larger were given a double cage (a portal was opened between two adjoining cages) per state standards. Cats were fed to maintain a stable body weight (200–225 kcal/cat/day) with about a tablespoon of canned food fed twice daily after each training session (approx. 37 kcal) and ¼ cup dry food at 6 pm (approx. 102 kcal), together with approximately 25–50 kcal of treats in training sessions.

Each cat underwent two, five-min clicker training sessions per day at approximately 11:00 a.m. and 4:00 p.m. Monday through Thursday each week (except on the first Monday when they were trained only once). This resulted in a total of 75 min of training time per cat over the two-week period. Training took place in the cage, in a small storage room, or in the main room of the facility, depending on the cat’s level of comfort and ability to handle increasing distractions, as well as whether visitors or volunteers were present and handling other cats at the time. The same two trainers attempted to teach each cat four behaviors: target, spin, sit, and high-five (purposeful hand contact with a paw). Trainer One worked with the cats in the morning session and Trainer Two in the afternoon session. Descriptions and criteria for successful task completion are shown in Table 1. For the pre-assessments and post-assessments, both verbal cues and visual cues (a target—either a finger/hand or a plastic chopstick) were used to elicit a behavior (no food lures were used). A clicker training manual that described suggested training protocols for each behavior (written by one of the study authors [22] and available upon request) was used as a reference for training. All cats were admitted to the facility on Fridays and allowed to acclimate over the weekend. Training began on the following Monday afternoon. In addition, each cat received at least ten minutes daily of one-on-one interaction with a person, such as petting or interactive play, unless the cat did not want to participate. These interactions took place either in the cage or in one of the facility’s rooms depending on the cat’s level of comfort and what else might have been going on in the facility at the time (visitors, other cats out, etc.). All cats were offered variable-length, group or individual out-of-cage time each day while cages were being cleaned, staff was doing other work, and/or during staff lunchtime or overnight.

At the beginning of each cat’s pre-assessment, a simple food preference test was administered whereby the cat was offered, at the same time, a small amount of chicken baby food on one plastic lid and a small amount of canned tuna on another plastic lid. The cat was given one min to interact with the food; interest in food was scored on a scale of 0–3, as illustrated in Table 2, below. For assessment purposes, interest in food levels were combined to create a high food interest (levels 2 and 3) and a low/no food interest (levels 0 and 1).

Once the cat’s preferred food choice was established, it was offered as a reward during the clicker training sessions. If the cat did not chose a food, then during the first training session, the trainer offered a variety of foods (including chicken baby food, canned tuna, canned cat food, dry cat food, and hard or soft commercial cat treats) one at a time until the cat ate. If the cat still did not eat, this process was repeated at subsequent training sessions until the cat ate. In rare cases, the cat remained uninterested in food for training sessions but enjoyed petting; in this case, petting was then offered as the reinforcer. The cats started training with the initial food selection, but if they lost or had no interest in the food, alternate choices were offered.

During the pre-assessment, each cat was rewarded with the treat they appeared most interested in eating based on the food preference test. Alternatively, if the cat did not display a preference (did not choose either food), the trainer arbitrarily chose the food to be offered as the reinforcer during the pre-assessment. Together, this resulted in 38% being offered chicken baby food and 62% being offered canned tuna.

The pre-training assessments and post-training assessments took place with the cat out of its cage (in the small storage room of the CLICK facility) unless the cat was too fearful to leave its cage, in which case assessments were performed with the cat in the cage. Cats were categorized as “shy” or “bold” during both pre-assessments and post-assessments. A cat was labeled “shy” if it was unwilling to leave its cage (either on its own when invited to, or if the trainer believed picking it up would compromise the cat’s welfare because it was displaying fearful or aggressive behaviors). If the cat easily came out of its cage, it was labeled “bold.” In the pre-assessments, cats that were too frightened of the assessor or of the cue or those that were positioned such that behaviors could not be cued correctly were not asked to perform all behaviors. Thus, the number of cats tested for each behavior in the pre-assessment does not always equal 100 and variances are noted. (This was not an issue for the post-assessments).

Assessments were comprised of Trainer Two cueing the cat (with both verbal and visual cues) to perform each of the four behaviors (target, sit, spin, and high-five) five times in a row. All pre-assessments and post-assessments were videotaped and analyzed by two independent reviewers who rated the cats’ behaviors on a 4-point scale from 0 (no semblance of the cued behavior performed) to 3 (complete mastery of the cued behavior-defined as performing the full and final behavior precisely within two seconds of being presented with the cue) (Table 3). After reviewers were thoroughly trained, they watched the videos as many times as needed and assessed the cats’ behaviors. Prior to discussion with each other, interrater reliability, measured with Cronbach’s Alpha, was 0.990. After individually scoring the behaviors, the reviewers met together to discuss any differences in their ratings. At this time, they watched the videos again if needed to reach a consensus. Reviewers were aware of the study’s goals and knew which videos were pre-assessments and which were post-assessments.

The data was entered into Excel and analysis was conducted with the statistical software SPSS (version 23) (IBM Corp, Armonk, NY, USA) Basic frequencies are reported and Wilcoxon signed-rank test was used to determine significant differences between the median scores of the pre-assessments and the post-assessments for each cat’s four scored behaviors. This was determined by calculating the change in scores between the pre-assessment median score and the post-assessment median score for each cat for each behavior. This delta targeting was what was used to determine change from pre-assessment to post-assessment.

The ages of the cats ranged from 6 months to 12 years (mean 3.55 years (SD 2.58), median 3 years). To analyze the impact of age, this variable was divided into 3 groups (young 0.5–2 years old), middle age (2.5–6 years old), and older/senior (7 years and older), and the Kruskal-Wallis test was used to assess the impact of age on trainability. The Mann-Whitney U test was used to assess the impact of sex, food interest, and initial shyness on trainability. Each of the four behaviors cued in the pre-assessments and post-assessments were scored, regardless of the cats’ performances. Median scores were calculated based on all five attempts for each behavior.

## 3. Results

### 3.1. Scores for Taught Behaviors

#### 3.1.1. Target

Using the Wilcoxon signed-rank test, the difference between the median scores of pre-assessment targeting behaviors and post-assessment targeting behaviors (touching a plastic chopstick or finger with nose; *n* = 100 in both pre-assessment and post-assessment) were used for analyses and found to be significantly different (*p* < 0.001). Forty-three (43%) cats showed a positive change between pre-assessments and post-assessments, nine (9%) showed a negative change, and 48 (48%) demonstrated no change. The pre-assessment for touching a target found that 48 cats (48%) scored a 3 (full nose contact with the target), compared with 79 (79%) at post-assessment.

#### 3.1.2. Spin

Evaluating the differences between pre-assessment spinning behaviors and post-assessment spinning behaviors (pre-assessed spin, *n* = 89; post-assessed spin, *n* = 100), a significant difference was found (*p* < 0.001), whereby 52 (58.4%) cats showed a positive change between pre-assessments and post-assessments, 11 (12.4%) showed a negative change, and 26 (29.2%) had no change. In the pre-assessments of spinning, 23 (25.8%) cats scored a 3, compared with 60 (60%) in the post-assessments.

#### 3.1.3. Sit

A similar change was found for sitting behavior (pre-assessed sit, *n* = 73; post-assessed sit, *n* = 100), with a significant difference found between pre-assessments and post-assessments (*p* = 0.001). Thirty (41.1%) cats demonstrated a positive change, 12 (16.4%) showed a negative change, and 31 (42.5%) had no change. Pre-assessments of sitting showed that seven cats (9.6%) scored a 3, compared with 27 cats (27%) in the post-assessments.

#### 3.1.4. High-Five

When assessing high-five behaviors (pre-assessed high-five, *n* = 86; post-assessed high-five, *n* = 100), the Wilcoxon signed-rank was used to test for a significant difference between pre-assessments and post-assessments (*p* < 0.001), and 43 (50%) cats demonstrated a positive change, 0 had a negative change, and 43 (50%) showed no change. Pre-assessments of high-five showed that no cats scored a 3, compared with 31 cats (31%) at post-assessment.

#### 3.1.5. Sex and Age

The effect of cats’ sex was assessed using the Mann-Whitney U Test and found to have no effect on the differences in pre-median scores and post-median scores for target (*p* = 0.52), spin (*p* = 0.64), sit (*p* = 0.46), and high-five (*p* = 0.06). Additionally, using the Kruskal-Wallis test, no significant differences in pre-median scores and post-median scores were found based on age (target, *p* = 0.44; spin, *p* = 0.51; sit, *p* = 0.95; high-five, *p* = 0.94).

### 3.2. Learned Behaviors and Food Interest

A Mann-Whitney U test was performed for each behavior (target, spin, sit, and high-five), by calculating the delta between the pre-assessment and post-assessment for each behavior, to test the hypothesis that cats with higher levels of food interest would demonstrate greater improvement in the four behaviors. Two behaviors that had significant pre-assessment and post-assessment differences based on food motivation were high-five (*p* = 0.001) and target (*p* = 0.018). Pre-assessment and post-assessment differences for the behaviors of spin (*p* = 0.10) and sit (*p* = 0.96) were not significantly different based on food interest (Table 4).

### 3.3. Learned Behaviors and Shyness

To test the hypothesis that a cat’s initial level of shyness could be used to help identify cats most likely to learn behaviors through clicker training, the Mann-Whitney U test was used to determine the association of the pre-assessment location on behavior changes by calculating the delta target for each behavior. For this purpose, pre-assessment location groups were labeled “in cage” (25 cats, 25%) or “out of cage” (75 cats, 75%). Although the number of cats in each category was the same for the pre-assessment and the post-assessment, they were not actually all the same cats. Some cats that were labeled “in cage” at pre-assessment were able to leave their cages for post-assessment, while other cats that were able to leave their cages during pre-assessment were not able to be tested outside the cage at post-assessment. No statistical differences were found for any behaviors. When the Mann-Whitney U test was used to examine the delta target between the post-assessment location for an association with behavior changes (“in cage” (25 cats, 25%) or “out of cage” (75 cats, 75%)), all behaviors were significantly different. There were higher median differences between pre-assessment and post-assessment for each behavior for those cats able to be tested outside their cage (target, *p* = 0.002; spin, *p* < 0.001; sit, *p* = 0.004; and high-five, *p* = 0.032).

## 4. Discussion

The results of this study suggest that it is possible to clicker train cats to perform specific tasks. In our sample of 100 cats, after a two-week training period that included 15, five-min training sessions, 79% of cats had mastered the ability to target (touch their nose to a plastic chopstick or finger), 27% mastered the ability to sit, 60% the ability to spin, and 31% the ability to high-five. This does not include the cats who may have come close to mastering the behavior (score of 2). For instance, a cat may have repeatedly been within 0.5 cm of touching its nose to the target, but if no actual contact was registered, a score of 3 was denied. For sitting, many cats did not actually touch the floor with their tailbones and therefore scored a 2 even if they came quite close to touching the floor. If we combine the behavior scores of 2 and 3 for targeting, then 95% of cats performed well; for sitting, this percent was 59%. For spinning, 76% of the cats performed well with behavior scores of 2 or 3 and for high-five, 38% had scores of 2 or 3.

Some cats demonstrated mastery of one or more of these behaviors at the pre-assessment, but the majority of those that demonstrated mastery post-assessment did so only after training. For example, at pre-assessment, only 48% of cats demonstrated mastery of targeting (the most commonly performed behavior at pre-assessment), 25.8% demonstrated mastery of spinning, and 9.6% demonstrated mastery of sitting on cue. No cats demonstrated the ability to high-five prior to training. Therefore, the hypothesis that cats can be clicker trained in a shelter environment was supported. In addition to lack of training, however, other factors that may have resulted in a cat’s poor performance during the pre-assessment include: (1) slight differences in the trainer’s cues for each cat, based on where the cat was positioned and the variability of normal human behavior, and (2) the cat’s level of stress or distraction because the environment and the trainer were unfamiliar. It is important to note that being housed in a shelter is likely one of the most stressful living arrangements these cats have ever encountered, and yet they were still able to learn new behaviors over the course of two weeks.

The hypothesis regarding a priori identification of cat demographic characteristics associated with successful learning was not supported; age and sex did not affect performance. While few people might think sex can play a role in trainability, it seems a relatively common public notion that older animals may not learn new behaviors as well as their younger counterparts (i.e., “you can’t teach an old dog new tricks”). In this study, that idea was not supported.

There was some support for the hypothesis regarding interest in food and successful learning. Initial interest in food did have an impact on two behaviors (high-five and target) yet it did not affect mastery of spin or sit behaviors. For example, a high percentage (72.9%) of cats that did not appear interested in food in the pre-assessment still mastered targeting, compared to 87.8% of food-motivated cats. This improvement was seen for each behavior, whereby numerous cats that appeared initially unmotivated by food successfully learned the behavior. The percentage of cats who learned to master sitting behavior, for example, was nearly the same for those that were food motivated (26.8%) and those were not food motivated (27.1%). However, many cats who were not food motivated in the pre-assessment became more interested in food during subsequent training. This improvement in appetite may be a result of decreased stress levels as the cats acclimated to the CLICK environment and the trainers, and/or because the cats were offered different food rewards that proved more motivating for those individuals. In addition, a small number of cats never became interested in food during training but found petting rewarding instead, and with this reinforcer, they learned at least to target.

Similarly, the hypothesis related to shyness and trainability was only partly supported. Initial shyness, assessed by a cat’s willingness to leave the cage for training, was not a significant predictor of trainability. Stated differently, there was no relationship between a cat’s ultimate mastery of a behavior and its initial shyness level. What was significantly related to training was how shy a cat was at post-assessment. Cats that had adapted to the point of being able to willingly leave their cage for assessment were more likely to demonstrate mastery for one or more of the four behaviors. Although some of the cats were shy (would not comfortably leave their cage) at both pre-assessment and post-assessment, 44% of initially shy cats adapted and were able to leave their cages for post-assessment. This is in agreement with Slater et al. [23] who noted that even over the course of just three days, regular interaction with a person can result in an increase in friendly behaviors in a shelter cat [23]. Together, these studies suggest that human-cat social interaction (including clicker training) can have a positive effect on the outcome (adopted or euthanized) for a cat in a shelter. For instance, such interactions can encourage a cat to come to the front of its cage when a person approaches, making them more visible to potential adopters. Also, these positive interactions can help decrease stress, thus lowering the cat’s risk for stress-related illnesses and for the display of fear-aggressive behaviors, both of which might otherwise result in euthanasia.

It is suggested, therefore, that using sex, interest in food, or initial shyness as exclusion criteria for clicker training is not supported. Results of this study suggest that many cats, regardless of these factors, are able to master new behaviors. In addition to its usefulness in training cats to perform cued behaviors, clicker training can also positively affect cat welfare and potentially increase adoption rates [3]. Although some cats adapt relatively well to confinement, other cats find confinement stressful, resulting in behavioral and physical problems [24,25,26]. Boredom and the inability to engage in natural behaviors during confinement can lead to the development of chronic stress and can have a negative impact on shelter cats’ well-being, which in turn can affect whether they have a live outcome [3].

Several forms of environmental enrichment, including both social and inanimate enrichment, have been used to combat stress and improve welfare. Although individual cats will respond differently to any type of enrichment, it has been suggested that toys and feeding enrichment such as puzzle feeders may be helpful [4]. For cats in particular, it has been found that letting them out of their cages and providing them with a stimulating activity is beneficial to their physical health and mental well-being [3,27].

In addition to providing cognitive stimulation, clicker training might also improve cats’ adoptability. This is another important factor in their welfare, because an extended shelter stay impacts physical and mental well-being in dogs and cats [28,29,30]. One factor that appears to positively influence likelihood of adoption is perceived friendliness [31], so anything that can increase an animal’s friendliness toward humans should be encouraged [32]. Positive human contact has been found to increase approach behavior and decrease fear in dogs [33] and cats [28]. Environmental factors that reduce cats’ general stress level also increase their approach behavior toward humans [34]. Additionally, Luescher and Medlock [35] found that obedience training improved dogs’ chances of adoption. Clicker training provides obedience training and socialization time with humans, both of which positively influence adoptability [36]. So, even though physical characteristics cannot be manipulated to make a cat more desirable (as deemed by potential adopters) [37], as demonstrated in the current study, desirable behaviors can be achieved through training.

Limitations of the current study include the fact that only one shelter environment was used as the testing facility, so generalizations should be made with caution. Additional studies in other shelters are warranted to ensure replicability of results. Future studies should also include assessment of additional tasks, including those deemed as most important to potential feline adopters. These might include such things as coming to the front of the cage, being willing to be held, or playing with a toy. It will also be important to assess the cats that did not change their behaviors to help determine contributing factors. Additionally, future studies exploring the use of clicker training to help facilitate veterinary visits by training cats to enter cat carriers should be explored as one way to help promote feline health.

## 5. Conclusions

In conclusion, this study supports the premise that cats can be effectively clicker trained to perform a variety of tasks in a relatively short period in a shelter environment. From this, it follows that owned cats that already have an established bond with their owner have great potential for being clicker trained. This training has the potential to modify unwanted behaviors and enhance the human-animal bond, and both of these factors can reduce the likelihood of relinquishment. These conclusions carry important welfare implications and warrant additional study.

## Figures and Tables

**Table 1 animals-07-00073-t001:** Behavior scoring parameters for cats’ behavior when cued. In all cases, if no semblance of the cued behavior was offered, the score was 0.

Behaviors	Scores		
	1	2	3
Target: Cat touches a plastic chopstick or trainer’s finger with its nose	Stretching neck toward target	Purposeful movement towards target, but no touch; or touches with face but not nose	Actual contact with nose
Spin (in either direction): Cat turns body in a circle	Head follows or <90 degree turn	Spins approximately 91–270 degrees	Spins approximately 271–360 degrees
Sit: Cat assumes a sitting position, with hind-end on floor and front paws touching the floor, in a weight-bearing stance	Some hind-end crouching	Hind-end on floor or in close proximity and/or front paws go up	Front paws on floor and hind-end contact with floor
High-five: Cat touches a trainer’s hand with a front paw	Any degree of front paw lift (one or both paws)	Movement towards hand with just one paw	Purposeful hand contact with one paw

**Table 2 animals-07-00073-t002:** Scoring system for interest in food upon presentation during pre-assessment.

Scores			
0	1	2	3
Did not investigate any food/ignoring food	Barely interested in food; sniffed at it but did not eat/mostly ignored food	Somewhat interested in food; eventually chose a food but not immediately	Very interested in food; chose and ate a food almost immediately

**Table 3 animals-07-00073-t003:** Comparison of median pre-assessment and post-assessment scores for four taught behaviors in 100 cats *.

Behavio	Pre-Assessment Scores					Post-Assessment Scores				
	*n*	0	1	2	3	*n*	0	1	2	3
**Target ***	100	30 30%	3 3%	19 19%	48 48%	100	2 2%	3 3%	16 16%	79 79%
**Spin ***	89	24 27%	18 20.2%	24 27.0%	23 25.8%	100	12 12%	12 12%	16 16%	60 60%
**Sit ***	73	34 46.6%	12 16.4%	20 27.4%	7 9.6%	100	31 31%	10 10%	32 32%	27 27%
**High-Five ***	86	85 98.8	1 1.2%	--	--	100	51 51.0%	11 11.0%	7 7.0%	31 31.0%

* *p* < 0.001; Note: In the pre-assessments, cats that were too frightened of the assessor or of the cue or those that were positioned such that behaviors could not be cued correctly were not asked to perform all behaviors.

**Table 4 animals-07-00073-t004:** Post-assessment median behavior scores and relation to food interest.

Behavior Score	0	1	2	3
**Target ***				
Little/no food interest (*n* = 59)	2 (3.4%)	2 (3.4%)	12 (20.3%)	43 (72.9%)
High food interest (*n* = 41)	--	1 (2.4%)	4 (9.8%)	36 (87.8%)
**Spin**				
Little/no food interest (*n* = 59)	8 (13.6%)	9 (15.3%)	14 (23.7%)	28 (47.5%)
High food interest (*n* = 41)	4 (9.8%)	3 (7.3%)	2 (4.9%)	32 (78.0%)
**Sit**				
Little/no food interest (*n* = 59)	24 (40.7%)	5 (8.5%)	14 (23.7%)	16 (27.1%)
High food interest (*n* = 41)	7 (17.1%)	5 (12.2%)	18 (43.9%)	11 (26.8%)
**High-five ****				
Little/no food interest (*n* = 59)	40 (67.8%)	5 (8.5%)	1 (1.7%)	13 (22.0%)
High food interest (*n* = 41)	11 (26.8%)	6 (14.6%)	6 (14.6%)	18 (43.9%)

* *p* = 0.018, ** *p* = 0.001.

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
