# Peer review of "Assessment of Clicker Training for Shelter Cats"

_animals, 2017, doi:10.3390/ani7100073_

Round 1

Reviewer 1 Report

This paper aimed to investigate whether clicker training can successfully be used with individually cage-housed shelter cats. The main results are that cats can be successfully trained in 4 tasks and that neither age nor sex are related to training success. Factors found to potentially influence training success where interest in food and personality of the cat.

This paper is in many parts well written and contributes to the existing literature on welfare and adoptability of shelter cats. Although there are many books on the use of clicker training in cats for lay people and clicker training is recommend for cognitive enrichment and behavioural modification of confined cats there are only a very low number of studies that investigated the use of (clicker) training in cats e.g. welfare/health (Gourkow, Nadine, and Clive JC Phillips. "Effect of cognitive enrichment on behavior, mucosal immunity and upper respiratory disease of shelter cats rated as frustrated on arrival." Preventive veterinary medicine 131 (2016): 103-110.).    

However, there are some aspects in particular in the methods and results section that should be supplemented or clarified.

Summary and abstract are fine

The introduction section is well written but is quite long presenting some aspects such as evolution of the cat, number of cats and money spent on cats in the US in more detail than needed for this research question. The introduction should be more concise in my view.

The methods section needs some additions and clarifications:

Line 132-133: how many cats had to be removed from the study?

Line 135: Please add details on the cats housing: e.g. cage size, which elements contained the cage? E.g. hiding box, litter box, toys, opportunity to scratch etc.; details on feeding and care of the cats

Line 137: where did the training take place and when (morning/afternoon, time between two sessions)?

Line 138: Was an individual cat always trained by the same trainer or did both trainers train all cats?

e.g. Line 139: Which cues where used und was food used to lure the cats, in particular in the pre-assessment? It would really be useful to provide the training manual in the supplementary materials section.

Line 144-146: where did the interaction with a person take place and where was the cat allowed out-of-cage time and was a human always present during this time?

Line 151-157: please add more details about preference for food testing: when prior to training was the food preference test carried out? How many times was a particular food presented to the cat and what was the order of the food presentation. How many times was the cat given to react to a particular food. How was the preferred food choice established e.g. by median scores? Did all cats have a clear preference? It would be interesting to present the data of the food preference testing.

Line 158-161: where did the pre- and post-assessments take place if not in the cage of the cat? Was the trainer doing the pre- and post-assessments one of the trainers involved in later training or another person?

Line 162-163: where the reviewers blinded to the study question and whether the video was a pre- or a post-assessment? Did you calculate an inter-rater reliability before discussion between the two raters?

Further additions: The way you categorized shy vs. bold should be added to the methods section.

Line 168-172: You used Mann-Whitney-U-tests to assess impact of age (besides, how was age categorized? young/old?) and other factors. Did you use only post-assessment median scores or did you calculate the difference/delta between the median pre-assessment score and the median post-assessment score for testing for effects e.g. of food preference? The latter would in my view be more appropriate as it represents the change over time. Please describe data analyses in more detail in the methods section.

Figure 1 and Figure 2 have the content of tables and my preference would be that they are presented as such.

The results section needs some additions and clarifications:

175-176: can you also provide length of stay in the shelter prior to training and source/intake mode of the cats e.g. stray, owner surrender? I assume that all cats were neutered? Please nevertheless add this information.

Line 180-181: The sentence about alternative foods should be placed in the methods section and the alternative foods should be listed as well as other potential rewards used e.g. petting.

Line 183-184: These two sentences would in my view also better fit in the methods / data analyses section.

Line 184-186: I think there is something wrong with this sentence!

Table 1: It would help to include an additional column with the total number of cats tested in the pre-assessment to have a quick overview.

Line 196-220: Please include the proportion of cats in the description of the changes (positive, negative, no change) between pre- and post-assessment.

Line 224-225: which scores of the initial food interest score are included in the high and low food interest group? Please describe this in the methods / data analyses section.

Line 234-235 and Line 237-238: pre-assessment location and post assessment location numbers are the same: both 25% and 75%? Is this correct? (Probably not after having read the discussion) Are those the same or other cats?

Line 237: is behavior change a calculated difference between pre- and post-assessment score or only the post-assessment score, same question for data in Table 2? See also my comment to line 168-172 in the methods comments part

Discussion:

Line 253-257: Mastery of behaviors in the pre-assessment might depend on trainer behavior and use of lures – please discuss whether for example cues for behavior or food used to lure cats might have resulted in that.

Line 262-263: This seems to be not consistent with the hypothesis and expectations formulated at the end of the introduction section?

Line 267-274: Where other rewards used during training? Such as attention through gently speaking to cat, petting/stroking, playing with the cat? I assume that there was some reward present so that also the less food motivated cats learned some of the behaviors. Alternatively stress during the food preference test might have inhibited interest in food – how long were the cats in the shelter and in the new housing in the CLICK-facility before the food preference test? Was food motivation related to shyness? This would support the assumption that low interest in food could be a result of stress rather than a generally low food motivation.

Line 285-286: You could give more detail about how human-cat interaction might mediate a positive effect on the outcome e.g. which effects of interactions that could be shown in other studies are positive for the outcome?

Line 290-291: As you did not test whether clicker training increases welfare of adoption rates it would be useful to cite literature that supports your statement.

Line 300-301: Reference 31 is a paper on cats not several species, furthermore this paper describes a protocol but did not test it for effectiveness; please add references about other species and trials that found effectiveness of training to reduce stress to support you statement.

Line 323-326: This statement is not consistent with the suggestion in Line 287-289?

Some general conculding remarks: What about the cats that did not change their performance during training? Did training success/failure depend more on the individual cat or the task? I think that including a control group without training would have strengthened the results of the study, however in my view it seems very unlikely that cats improve their skills markedly in those task without any training. These aspects should be included in the discussion.

Author Response

Thank you,

Lori Kogan

Reviewer 2 Report

The premise is excellent but if the goal is to increase adoptability and long term homing, is spinning a reasonable behavior. I'm guessing you chose it for being a truly novel behavior.  If homing is the ultimate goal perhaps come or play now would be better behaviors to try to instill.

Line 32 I think cat's is the correct term because multiple cats don't have a collective sex or age

Line 68 Do these problems occur because cats are in shelters? Statement seems to imply they do but neither cited study was in a shelter situation.

Line 122 I think temperament is preferred over personality by behaviorists

Line 252 Only 38 has no % sign beside it; you may want to be consistent

Line 291 Again as in line 68 confinement intimates shelter situations which the article does not use as a source of information

Line 305 I would suggest replacing since with because.  Since implies a passage of time which is not occurring her but rather cause and effect which because indicates. See book Eats Chutes and Leaves as reference.

Author Response

Thank you,

Lori Kogan

Round 2

Reviewer 1 Report

Thank you for revising the manuscript. I still have some objections – in particular to the results section:

Methods section:

Please include the use of the Kruskal-Wallis Test and the categorization of age in the data analyses part (Line 201-207).

Results section:

If possible please include descriptive statistics for length of stay and source of the cats (Line 214- 216).

Line 219 – 221: What is meant by initial food? Further this statement about cats with no preference is not consistent with the description in the methods part (Line 176-178).

Line 223-224: It is not possible to include sex as a variable in a Wilcoxon signed-rank test. The most appropriate way to test for sex differences in trainability would be to calculate the change in score between pre- and post-assessment by subtracting the pre-assessment median score from the post-assessment median score for each cat and each behavior (e.g. 3-1=2 that means an improvement of 2 scores in this cat) and then use this calculated difference as the variable (e.g. called “delta targeting” which represents the change from pre- to post-assessment)  tested against sex in a Mann-Whitney U Test. Same way for location in cage and food interest. For age a Kruskal-Wallis test can be used as you reported. However, your description of how you did your statistics is still not that clear and I assume from the wording that you tested both pre- and post-assessment median scores with a Kruskal-Wallis test?  Even if not significant p-values should be reported.

Line 235-237: The changes made in this sentence are not appropriate as the Wilcoxon signed-rank test does not use a calculated delta (as described above) but the pre- and post–assessment scores. Please return to the former version.

Line 262 – 264: Did you perform the Mann-Whitney U-Test on a calculated delta as described above or on the pre- and post-assessment scores? Table 4 represents the post-assessment scores I assume? If the pre- and post-assessment scores were used, than please make this clear in the text and the table caption. Only using a calculated delta represents change or ev. improvement. The pre- or post assessment scores represent only one point in time and do give us no information about the cats progress if used as a single variable. E.g. in Table 4 the caption should say “Post-assessment median behavior score….” unless it represents the caculated difference/delta as described above. Looking at the other descriptive data I assume it does not represent the change but the post-assessment scores?

Line 280-282: delta = behavior change; This sentence is confusing. Same problem as above. It is not clear whether a calculated delta or just pre- or/and post-assessment scores were used.

Discussion section:

Line 350: References – there is one bracket sign too much at the end.

Thank you for keeping up!

Author Response

Reviewer -Round 2-Minor

Thank you for revising the manuscript. I still have some objections – in particular to the results section:

Thank you for allowing us to strengthen the paper. We are hopeful we have clarified things; in particular, the fact that we did use the delta for analysis when appropriate.

Methods section:

Please include the use of the Kruskal-Wallis Test and the categorization of age in the data analyses part (Line 201-207).

This has been modified: “The ages of the cats ranged from 6 months to 12 years (mean 3.55 years [SD 2.58], median 3 years). To analyze the impact of age, this variable was divided into 3 groups (young 0.5 – 2 years old, middle age (2.5 – 6 years old) and older/senior (7 years and older)) and the Kruskal-Wallis test was used to assess the impact of age on trainability.”

Results section:

If possible please include descriptive statistics for length of stay and source of the cats (Line 214- 216).

            I am sorry, these data are not available.

Line 219 – 221: What is meant by initial food? Further this statement about cats with no preference is not consistent with the description in the methods part (Line 176-178).

This has been clarified: “During the pre-assessment, each cat was rewarded with the treat they appeared most interested in eating based on the food preference test. Alternatively, if the cat did not display a preference (did not choose either food), the trainer arbitrarily chose the food to be offered as the reinforcer during the pre-assessment. Together, this resulted in 38% being offered chicken baby food and 62% being offered canned tuna.”

Line 223-224: It is not possible to include sex as a variable in a Wilcoxon signed-rank test. The most appropriate way to test for sex differences in trainability would be to calculate the change in score between pre- and post-assessment by subtracting the pre-assessment median score from the post-assessment median score for each cat and each behavior (e.g. 3-1=2 that means an improvement of 2 scores in this cat) and then use this calculated difference as the variable (e.g. called “delta targeting” which represents the change from pre- to post-assessment) tested against sex in a Mann-Whitney U Test. Same way for location in cage and food interest.

This has been changed. The Mann-Whitney U Test was used to test effect of sex, as well as cage and food interest. We did use delta targeting. I have attempted to make this clearer throughout the results section.

For age a Kruskal-Wallis test can be used as you reported. However, your description of how you did your statistics is still not that clear and I assume from the wording that you tested both pre- and post-assessment median scores with a Kruskal-Wallis test? Even if not significant p-values should be reported.

            This has been clarified and non-significant p values have been added

Line 235-237: The changes made in this sentence are not appropriate as the Wilcoxon signed-rank test does not use a calculated delta (as described above) but the pre- and post–assessment scores. Please return to the former version.

This has been changed.

Line 262 – 264: Did you perform the Mann-Whitney U-Test on a calculated delta as described above or on the pre- and post-assessment scores? Table 4 represents the post-assessment scores I assume? If the pre- and post-assessment scores were used, than please make this clear in the text and the table caption. Only using a calculated delta represents change or ev. improvement. The pre- or post assessment scores represent only one point in time and do give us no information about the cats progress if used as a single variable. E.g. in Table 4 the caption should say “Post-assessment median behavior score….” unless it represents the calculated difference/delta as described above. Looking at the other descriptive data I assume it does not represent the change but the post-assessment scores?

Yes, the Mann-Whitney U-Test was used on a calculated delta. I have tried to make this clearer in the text. Table 4 caption has been changed to better reflect that is it post assessment median scores, not calculated differences.

Line 280-282: delta = behavior change; This sentence is confusing. Same problem as above. It is not clear whether a calculated delta or just pre- or/and post-assessment scores were used.

The Mann-Whitney U-Test was used on a calculated delta. I have tried to make this clearer in the text.

Discussion section: 2

Line 350: References – there is one bracket sign too much at the end.

This has been corrected.

Reviewer 2 Report

Excellent revision

Author Response

Thank you!

Lori Kogan

Animals EISSN 2076-2615 Published by MDPI AG, Basel, Switzerland RSS E-Mail Table of Contents Alert
Back to Top